# Generating high-quality plant and fish reference genomes from field-collected specimens by optimizing preservation

Jeremiah J. Minich[1,3], Malia L. Moore [1,2,3], Nicholas A. Allsing[1], Anthony Aylward[1], Emily R. Murray[1], Loi Tran[1] & Todd P. Michael [1✉]

Sample preservation often impedes efforts to generate high-quality reference genomes or pangenomes for Earth's more than 2 million plant and animal species due to nucleotide degradation. Here we compare the impacts of storage methods including solution type, temperature, and time on DNA quality and Oxford Nanopore long-read sequencing quality in 9 fish and 4 plant species. We show 95% ethanol largely protects against degradation for fish blood (22 °C, ≤6 weeks) and plant tissue (4 °C, ≤3 weeks). From this furthest storage timepoint, we assemble high-quality reference genomes of 3 fish and 2 plant species with contiguity (contig N50) and completeness (BUSCO) that achieve the Vertebrate Genome Project benchmarking standards. For epigenetic applications, we also report methylation frequency compared to liquid nitrogen control. The results presented here remove the necessity for cryogenic storage in many long read applications and provide a framework for future studies focused on sampling in remote locations, which may represent a large portion of the future sequencing of novel organisms.

[1] The Plant Molecular and Cellular Biology Laboratory, Salk Institute for Biological Studies, 10010 N. Torrey Pines Rd., La Jolla, CA 92037, USA. [2] Scripps Institution of Oceanography, University of California San Diego, 8622 Kennel Way, La Jolla, CA 92093, USA. [3]These authors contributed equally: Jeremiah J. Minich, Malia L. Moore. ✉email: tmichael@salk.edu

Long-read sequencing, the 2022 Nature method of the year, is enabling biologists to digitally archive genomes from bacteria to complex eukaryotes to understand fundamental questions in evolution, bolster conservation strategies in the midst of mass extinctions, discover the biosynthetic machinery behind pharmaceutically relevant specialized metabolites, and improve food production through crop and livestock genomics[1–3]. Large-scale initiatives such as The Earth Biogenome project and Darwin Tree of Life Project are employing this technology to generate reference genomes for all 1.8 million named eukaryotes on Earth including all 71,657 vertebrates by 2025[4–6].

Sample collection, storage and preservation are critical steps to ensure high molecular weight (HMW) DNA acquisition required for long-read sequencing. Standard methods, which include snap freezing in liquid nitrogen ($LN_2$) or on dry ice followed by storage at −80 °C, are often unrealistic in many low- and middle-income countries, particularly at remote locations. Alternative storage methods (solvent, buffer, desiccation) that are compatible with short-read sequencing have shown theoretical promise in long-read sequencing applications through analysis of DNA fragment sizes and purity[7]. However, these methods have yet to be benchmarked on a major long-read sequencing platform such as Oxford Nanopore Technologies (ONT) or Pacific Bioscience (PacBio), and have only been quality assessed at short storage times (hours) that are not realistic with travel times from remote regions[7]. Improving the capacity to collect, store, and transport samples by using readily available and cost-effective solutions is critical to both the improvement of crop production through pangenomic efforts and the successful generation of high-quality reference genomes for all life on Earth.

Here we evaluate the impacts of storage solutions (95% ethanol [EtOH] and RNAlater), temperature (4 °C and 22 °C), and time (0 days, 4 hours, 2 days, 1 week, 3 weeks, and 6 weeks), on HMW DNA quality, ONT sequencing quality (LSK114, R 10.4.1) and assembly quality. We sampled fish ($n = 9$ species, 90 samples) and plants ($n = 4$ species, 36 samples) to establish the protocols across the tree of life. These complementary groups of organisms are realistically challenging to preserve in the field and represent different tissue types. We confirm the viability of solvent-preserved samples by producing high-quality de novo assemblies for five organisms at the furthest storage time point.

## Results and discussion

Fish were preserved in three distinct storage conditions: 95% EtOH at 4 °C, 95% EtOH at 22 °C, and RNAlater at 22 °C with $LN_2$ as a control, while plants were only preserved under two storage conditions due to constraints on tissue quantity: 95% EtOH at 4 °C and RNAlater at 22 °C with $LN_2$ as a control. A schematic of our experimental design is outlined in Fig. 1. A total of nine fish and four plant species were included in the analysis of storage conditions on DNA quality metrics (Supplementary Data 1). Seven plant samples from later storage time points were excluded from sequencing due to insufficient DNA (Supplementary Data 2). Comprehensive quality control (QC) and sequencing metrics are available in Supplementary Data 2, and statistical analysis across Supplementary Figs. 1–4. Source data is available in Supplementary Data 2.

First, we assessed the impact of storage conditions on DNA extraction yield by Qubit and purity by Nanodrop A260/280 and A260/230. All metrics remained stable for fish blood across all storage conditions through six weeks, with yields consistently greater than the liquid nitrogen control (Supplementary Fig. 1a–i). Greater variability was observed with plant tissue; EtOH 4 °C produced consistent yields as a function of time until week three, while RNAlater 22 °C produced higher yields at four

hours in storage buffer than the liquid nitrogen control, but decreased significantly across the remaining timepoints ($P = 0.0071$, Fs=11.40) (Supplementary Fig. 2a, b). DNA A260/230 purity was suboptimal in weeks one and three for plants (Supplementary Fig. 2c–f).

DNA fragment sizes measured by Femto Pulse produced two useful metrics: average DNA length and percentage of DNA greater than 50 kilobase pairs (kb). For fish, EtOH 4 °C performed 'best' with no differences compared to the control to six weeks. EtOH 22 °C had the greatest variation among time points for the fragmentation methods assessed, with average length differing from control with $P = 0.0216$, Fs = 9.667, and weeks three and six trending lower (Supplementary Fig. 1m–r). Plants retained DNA size with EtOH 4 °C up to three weeks but reduced dramatically in RNAlater 22 °C at later time points (Supplementary Fig. 2i–l). Interestingly, same-day four hour storage in RNAlater 22 °C yielded longer DNA than the control, which was not observed for EtOH 4 °C (Supplementary Fig. 2j, l).

Following DNA QC, 90 fish samples and 29 plant samples were barcoded and run at low coverage on ONT to determine sequencing quality by read quality and read N50 length. Statistical analysis was performed on the land plants only due to later timepoint marine plant exclusions, though the same trends are qualitatively evident in the larger dataset (Supplementary Fig. 2m–x). Encouragingly, read N50 length for fish did not significantly change with storage time in EtOH 4 °C and EtOH 22 °C, and was only negatively associated with extended storage times for RNAlater 22 °C (Supplementary Fig. 3a–c). Reduction in N50 length with storage time was significant in plants for both EtOH 4 °C ($P = 0.0417$, Fs = 7.600) and RNAlater 22 °C ($P = 0.0417$, Fs = 7.600) (Supplementary Fig. 2m–p). For read quality, all fish treatments were stable across timepoints, with EtOH 4 °C actually showing increased quality by week six (Supplementary Fig. 3d–f). Quality scores among the plant samples were stable for EtOH 4 °C treatment, but RNAlater 22 °C showed reduced quality by day two (Supplementary Fig. 2q–t). DNA fragment sizes did not predict read N50 length for fish, whereas for plants, both DNA yield and fragment size estimates had a positive correlation with read N50 length (Supplementary Data 3). While we observed no association between sequencing yield and N50 length (Supplementary Fig. 4a), we found a significant correlation between library read quality and read N50 length for both fish and plants (Supplementary Fig. 4b). For both plant and fish biological replicates of different species, we did observe high variation in the sequencing read N50 length suggesting there are other factors at play (Supplementary Fig. 5a–f). Since field collected fish samples had additional variances associated with sampling method, we further investigated this as a factor. We did not find any significant association between delayed processing times of putting blood into EDTA tubes and sequencing N50 (Supplementary Fig. 5g) nor an effect of time on ice prior to transfer to storage solution (Supplementary Fig. 5h).

We were interested in whether size selection could rescue degraded samples and evaluated the performance of the ONT short fragment eliminator (SFE) kit on the fish dataset of 90 samples. The SFE kit had a positive impact on sequencing N50 ($P = 0.0443$) (Supplementary Fig. 6a). The N50 length increased in 63.3% of libraries by an average of 27%. The max increase was 318% (4092 bp to 17,094 bp) and 15.5% of libraries increased N50 length by at least 100% (Supplementary Fig. 6b). The application of the SFE kit had the greatest positive impact on highly fragmented libraries, whereas libraries that previously demonstrated a high sequencing N50 length were more negatively impacted by an SFE kit step (Supplementary Fig. 6c, d). Because of the overall positive effect, all DNA samples prepared for deep sequencing were processed with the SFE kit.

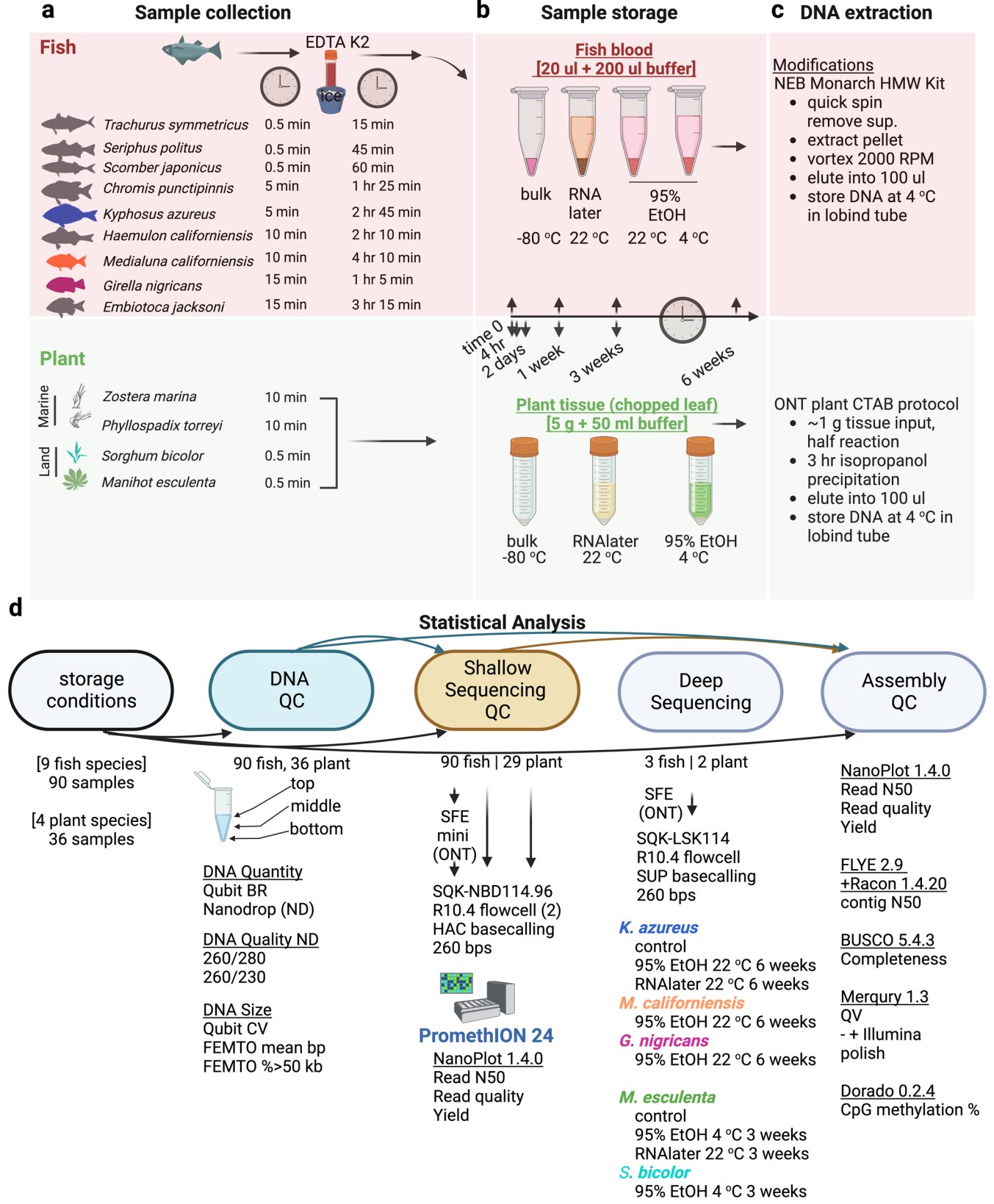

**Fig. 1 Experimental approach: comparing impacts of storage conditions on DNA QC, sequencing, and assembly metrics. a** Nine fish species and four plant species were collected and **b** preserved using various storage conditions for up to 6 weeks (fish) or 3 weeks (plants). **c** HMW DNA extraction methods with slight modifications. **d** Workflow for assessing impacts of storage conditions on DNA QC and shallow sequencing performance. A subset of species (indicated by color coding) at their storage extremes were fully sequenced and assembled. Figure was made using Biorender.

**Table 1 Sequencing and assembly results of fish and plant samples from extended storage conditions.**

| Spp. | Storage | | | Sequencing | | | | | Assembly | | | | | |
|---|---|---|---|---|---|---|---|---|---|---|---|---|---|---|
| | Solv | °C | wks | N50.s | N50.d | Long | Q_avg | cov(x) | N50 | contig | C(%) | QV.n | QV.hy | methy |
| Mcal | ET | 22 | 6 | 3.4 | 9.8 | 176.5 | 18.7 | 58 | 6.5 | 1761 | 98.7 | 36.2 | 37.9 | NaN |
| Gnig | ET | 22 | 6 | 3.5 | 15.2 | 139.4 | 17.9 | 37.8 | 9.7 | 1441 | 98.5 | 38.0 | 40.1 | NaN |
| Kazu | LN | −80 | 0 | 16.9 | 54.4 | 418.9 | 20.1 | 28.4 | 21.8 | 641 | 98.8 | 40.2 | 42.0 | 79.14 |
| Kazu | ET | 22 | 6 | 15.8 | 20.5 | 183.4 | 19.1 | 33.9 | 13.8 | 792 | 98.6 | 40.3 | 42.4 | 78.66 |
| Kazu | RN | 22 | 6 | 13.7 | 15.1 | 225.5 | 19 | 11.2 | 5.1 | 795 | 98.7 | 39.9 | 42.3 | 80.33 |
| Mesc | LN | −80 | 0 | 21.6 | 38.7 | 367.8 | 19.2 | 52.8 | 16.5 | 460 | 99 | 47.5 | 49.3 | 77.63 |
| Mesc | ET | 4 | 3 | 13.9 | 24.1 | 296.2 | 17.8 | 58 | 10.4 | 638 | 99.2 | 43.8 | 44.6 | 77.26 |
| Mesc | RN | 22 | 3 | 1.9 | 6.2 | 392.3 | 17.2 | 37.3 | 0.4 | 18881 | 99 | 32.1 | 32.5 | 75.55 |
| Sbic | ET | 4 | 3 | 9.1 | 12.7 | 394.9 | 16 | 82.9 | 5.9 | 1705 | 94.6 | 44.3 | 46.4 | NaN |

Spp: fish and plant species *M. californiensis* (Mcal), *G. nigericans* (Gnig), *K. azureus* (Kazu), *M. esculenta* (Mesc), *S. bicolor* (Sbic).
Solv: solvent used ET (95% EtOH), RN (RNAlater), LN (Liquid Nitrogen control).
N50.s: sequencing read N50 (kb) from shallow seq approach (NanoPlot) of untreated DNA.
N50.d: sequencing read N50 (kb) from deep seq approach (NanoPlot) of DNA processed through SRT.
Long: longest read observed in sequencing (kb).
Q_avg: mean quality score of library (NanoPlot).
cov(x): estimated sequencing coverage from ONT.
N50: the assembly N50 (Mb).
contig: total number of contigs from assembly.
C(%): BUSCO completeness.
QV determined using Merqury: QV.n from ONT only polishing with 'Racon' ; QV.hy from ONT and Illumina polishing with 'Pilon'.
methyl: % of Cytosine bases which are methylated as determined from ONT dorado basecaller.
NaN: refers to not performed.

Based on the promising shallow long-read sequencing results, we fully sequenced a subset of organisms at the most extreme conditions of time and/or temperature. Success at these extremes would suggest viability of intermediate samples from less extreme conditions. For five species, we deep sequenced and assembled genomes from the furthest storage time point: three previously un-sequenced species of fish (*Medialuna californiensis, Girella nigericans, Kyphosus azureus*) at EtOH 22 °C for six weeks, and the two terrestrial plants (*Sorghum bicolor, Manihot esculenta*) in EtOH 4 °C for three weeks. Comprehensive sequencing and assembly statistics are presented in Table 1. Notably, all EtOH stored samples assembled with contig N50 lengths greater than 6 Mb. *K. azureus* EtOH 22 °C assembled with a 13.82 Mb contig N50 length and 98.6% Benchmarking Universal Single-Copy Orthologs (BUSCO) complete genes (C:98.6%) from 33.9X coverage of the genome, comparable to the liquid nitrogen control with a 21.78 Mb contig N50 length and C:98.8% from 28.4X coverage. *M. esculenta* EtOH 4 °C sequence produced a 10.4 Mb N50 with C:99.2% from 58X coverage, while the control yielded 16.5 Mb N50 and C:99.0% from 52.8X coverage. Two included RNAlater 22 °C assemblies were reduced in quality compared to the EtOH: *K. azureus* 15.1 Mb N50 length, C:98.7% and *M. esculenta* 0.4 Mb N50 length, C:99.0%. Illumina reads associated with the same DNA samples were used to estimate the per base sequence quality (QV) using Merqury statistics to ascertain base-level accuracy among the assemblies both with and without Illumina polishing[8]. With solely ONT reads for assembly and polishing, five out of nine genomes met the minimum QV standards of >40 as proposed by the VGP. This suggests that a sufficient QV score is feasible with only ONT sequencing for higher-quality runs, but that this should be determined on a sample-to-sample basis. Illumina sequencing improved the genome QV scores in all nine genome assemblies by 0.4-2.3 or an average of 4.15% (S.D. 1.6), so that seven out of nine genomes QV > 40. The lower quality assemblies were *M. californiensis* (95% EtOH 6 weeks), which we hypothesize was due to this fish having the longest transit time before EtOH storage, and *M. esculenta* (RNAlater 3 weeks), which further demonstrates the incompatibility of extended RNAlater storage for a plant. These samples also had the lowest sequencing N50s. It may be feasible to still obtain high-quality genomes from these samples if additional sequencing is performed and the shorter read fragments excluded through filtering.

For *M. esculenta* and *K. azureus*, the additional liquid nitrogen controls allowed us to assess genome-wide frequency of CG methylation retained in solvent stored samples compared to snap-freezing. The fish *K. azureus* had a slightly lower methylation rate for the 95% EtOH storage as compared to the liquid nitrogen (LN$_2$) control (CV = 0.43%) whereas the RNAlater sample (CV = 1.08%) was slightly elevated Table 1. For the plant sample, *M. esculenta*, both the 95% EtOH (CV = 0.33%) and RNAlater (CV = 1.44%) samples were slightly lower than the LN2 control (Table 1). The coefficient of variation was higher in the RNAlater samples as compared to 95% EtOH stored samples, but overall, we find that sample storage in either 95% EtOH or RNAlater at both 4 °C and 22 °C has negligible impact on the ability to measure methylation profiles.

Both DNA fragment length and read N50 length from shallow sequencing predicted assembly outcomes from deep sequencing. DNA fragment size is positively correlated with deep sequencing read N50 length (P = 0.0141, R$^2$ = 0.6031) and resulting assembly contig N50 length (P = 0.0058, R$^2$ = 0.6860; linear model) (Fig. 2, Supplementary Data 5). Shallow sequencing N50 lengths is also positively correlated with deep sequencing read N50 length (P = 0.0174, R$^2$ = 0.5783) and assembly contig N50 length (P = 0.0220, R$^2$ = 0.5507) (Supplementary Fig. 7a, b). Of note, shallow sequencing N50 always underestimates deep sequencing N50 which we speculate is due to additional pipetting requirements for barcoding. Nonetheless, their correlation suggests that shallow sequencing can be a powerful QC method for evaluating sequencing suitability of large sample sets where methods like Femto Pulse are time-consuming or unavailable (Supplementary Fig. 7b). Genome coverage did not predict assembly N50 length, indicating that the range of sequencing depths represented in this dataset were more than sufficient (Supplementary Fig. 7d). The strongest predictor of assembly length was the deep sequencing read N50 (P = 0.0002, R$^2$ = 0.8709, linear model) (Supplementary Fig. 7c) although when read length was > 50 kb, other metrics such as the % of reads and % of bases were also highly predictive (Supplementary Data 4 and Supplementary Fig. 7e–l).

In conclusion, for optimal genome assembly results with long read sequencing, samples for DNA extraction should be stored at

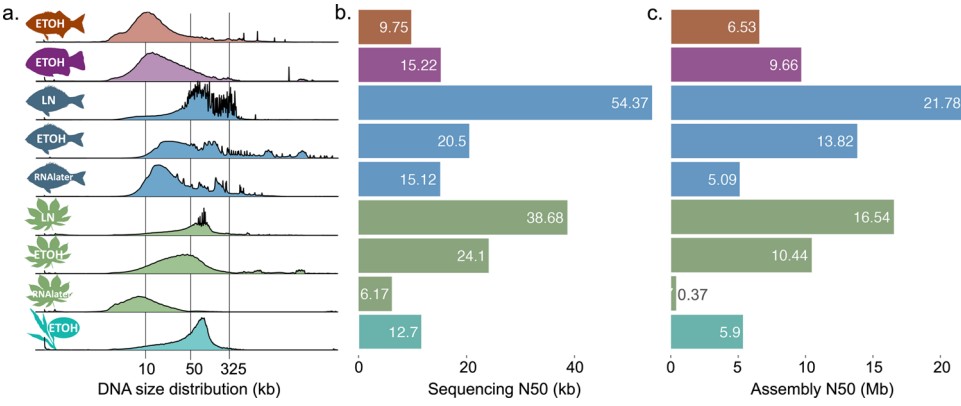

**Fig. 2 DNA fragment length predicted read length and assembly contiguity from deep sequencing. a** Femto Pulse DNA fragment length distributions, **b** deep sequencing read N50 length, and **c** assembly contig N50 length presented in the same sequence as Table 1: *M. californiensis* (orange), *G. nigericans* (purple), *K. azureus* (blue), *M. esculenta* (green), *S. bicolor* (turquoise).

the coldest temperatures and processed as soon as possible. However, here we show that 95% EtOH is a suitable storage buffer for both fish blood at 22 °C for up to 6 weeks and plant tissue at 4 °C for up to 3 weeks, as validated by high-quality assemblies (assembly N50 length > 1 Mb and BUSCO completeness >90%). Of the five assemblies from 95% EtOH stored samples, four achieved a per base quality QV > 40. An additional exploration of genome-wide CG methylation in *K. azureus* and *M. esculenta* found that for these organisms, sample storage had negligible impact on the ability to measure methylation profiles, especially with 95% EtOH (CV < 0.5% compared to $LN_2$ control). Overall, we recommend that for animals with nucleated blood, whole blood should be initially stored in an EDTA K2 tube on ice for up to four hours and transferred to 95% EtOH for up to six weeks at 22 °C but improved with refrigeration. RNAlater produces lower quality sequences with extended storage but may still generate high-quality assemblies for fish up to six weeks as an option in alcohol-restricted countries. Terrestrial plants may be stored in 95% EtOH and kept on ice or refrigerated up to three weeks or stabilized in RNAlater for same-day transport only. The quality reduction of the 1- and 3-week time points for marine plants is a caution to those working with marine or aquatic plants that have evolved barriers to liquid intrusion from living in an aqueous environment; they may be stored up to one week in EtOH at 4 °C and in RNAlater for same-day transport. Considering the great diversity of metabolic content within the plant kingdom, plant leaf storage time should be reduced whenever possible, and the potential of EtOH or RNAlater for storing more lignified plant tissues like stem or root for DNA extraction requires further study. For ascertaining sequencing suitability within large sample sets, we suggest that QC of DNA is optimally performed by directly sequencing through multiplexing and shallow sequencing on ONT. We've created a standard operating procedure (SOP) outlining instructions for collection, storage, and DNA extraction for both fish and plant samples (Supplementary Notes 1-2). We hope this work provides detailed guidelines for researchers working in remote areas to safely transport nucleated blood or plant leaves for genomic study, and that the success of these sample types demonstrated over a range of storage timepoints motivates others to test this simple method on other tissues.

## Methods

**Sample collection**. A total of nine species of marine fish were collected across three different sampling days (September 7th, 9th, and 12th 2022) under IACUC Animal Use Protocol S12219

(Supplementary Data 1). Six species were collected using a speargun donated by a local fisher. Fish were transported back to shore, euthanized, and blood extracted using a 22 gauge needle and syringe from the caudal vein. The range of transit times from when the fish was first collected until the blood was withdrawn was between 5 and 15 minutes. After blood was drawn, it was dispensed into an EDTA K2 tube and placed on ice. Tubes remained on ice between 1 hr and 5 minutes to 4 hr and 10 minutes for field-collected specimens. An additional three species of fish were collected from an experimental holding tank at Scripps Institution of Oceanography (SIO). These fish were initially collected from the SIO pier and housed in a tank for approximately one month. All three fish had a very short time from when the blood was collected until it was dispensed into the EDTA K2 tube (~30 seconds or less). These blood tubes also had the shortest time on ice until they were dispensed into the various storage buffers (15 min to 60 min). Fish ranged in size from 12.5 cm to 32 cm in total length and 14.1 g to 519.7 g total mass (Supplementary Data 1).

Plants were sourced on the same day (September 12, 2022) from various locations in San Diego, California: Cassava (*Manihot esculenta*) from the San Diego Botanical Gardens (SDBG), Sorghum (*Sorghum bicolor*) from Salk Institute for Biological Studies greenhouse facilities, eelgrass (*Zostera marina*) from Mission Bay, and surfgrass (*Phyllospadix torreyi*) from Windansea Beach (Supplementary Data 1). The eelgrass and surfgrass samples were collected by hand while snorkeling, with California Fish and Wildlife permit S-210200011-21023-001. To achieve relative consistency in tissue age and condition across treatments of the same plant, leaves were cut into 1 cm segments and aggregated before being allocated between storage conditions. For both cassava and sorghum, all samples came from the same plant. For the seagrasses, smaller plant size necessitated that three genetic individuals were collected, with one allocated to each storage condition to eliminate variability across timepoints. These individuals were all mature plants of comparable biomass.

**Sample storage**. Using a wide bore pipet tip, approximately 20 µl of fish whole blood kept in EDTA K2 tubes on ice was dispensed into 2 ml Eppendorf tubes containing either 200 µl of 95% EtOH or 200 µl of RNAlater. An additional 20 µl sample was dispensed into an empty 2 ml Eppendorf tube on dry ice that was then immediately stored at −80 °C and used as the control. For samples in buffers, 95% EtOH stored samples were stored at two temperatures of either 4 °C (representing refrigeration) or 22 °C, representing room temperature. The RNAlater stored samples

were only stored at 22 °C. Pilot experiments suggested that EtOH would have higher performance so we opted to include more temperature treatments with this buffer. We included RNAlater however, because some countries in the world restrict or prohibit all forms of EtOH thus alternative storage buffers are needed in those cases. Samples were then stored for 1 week, 3 weeks, and 6 weeks. At each time point, samples were immediately processed for DNA extraction.

At the time of plant tissue sampling, young leaf tissue was added to one of three treatments: $LN_2$, 95% EtOH on ice, or RNAlater at ambient temperature. For the EtOH and RNAlater, ~5 g tissue was added to 50 mL solvent in nonreactive glassware for storage, in accordance with the suggested RNAlater tissue to solvent ratio. EtOH samples were stored at 4 °C and RNAlater samples at room temperature in the dark to reduce oxidation. At each time point, ~1 g samples were removed from their solvent, blotted with KimWipes and flash frozen in liquid nitrogen for tissue grinding. The same-day time point (time 0.17) was captured 4 hours after collection, once samples reached the lab.

**Sample extraction**. Prior to extraction of fish samples stored in buffers, tubes were spun down for 2 minutes at 5000 rpm in a centrifuge. The storage solution supernatant was carefully removed with a pipet. Samples were then processed following the NEB Monarch HMW DNA extraction kit for cells and blood (New England Biolabs, Ipswich, MA, Cat#T3050L). For buffer stored samples, we followed the 'fresh nucleated blood' protocol. For the control samples which had frozen whole blood without buffer, we followed the 'frozen nucleated blood' protocol. During the lysis step, we used the highest recommended setting of 2000 RPM. For all fish samples we followed the 'standard input' for the various buffer formulations. In the end, we eluted with 100 μl of elution buffer. We deviated from the protocol in that we did not do any sort of pipet mixing as they recommend a 5-10x pipet mix to shear the DNA to better go into solution. We did this to maximize DNA length.

Plant samples were ground by mortar and pestle in liquid nitrogen to a fine powder and ~1 g of sample carried forward into extraction, estimated by a half teaspoon scoop. HMW DNA was extracted using the Oxford Nanopore plant extraction protocol for Arabidopsis (https://community.nanoporetech.com/extraction_method_groups/plant-leaf-gDNA), which uses components of the QIAGEN Blood and Cell Culture DNA Midi Kit (QIAGEN Cat#13343). This protocol was adapted for half-reactions in 20 mL lysis buffer to increase throughput by eliminating the combination of two lysed samples per column. Additionally, the final isopropanol precipitation was reduced to 3 hours for all extractions to accommodate samples in RNAlater. The experimental design of collection, storage, and processing is included in Fig. 1, which was made using BioRender with a paid license. Statistical analyses and figures made using either R or Prism version 9.4.1 with a paid license.

**DNA quality control**. All DNA was allowed to rest at 4 °C for at least one week before quantifying to ensure full solubilization of DNA. Endpoint measures were assessed for DNA QC spanning DNA yield, DNA purity, and DNA fragment size. DNA was quantified using both the Qubit (dsDNA quantitation) and Nanodrop. For the Qubit, 1 μl of DNA from the top, middle, and bottom of each tube was added to the Qubit Broad Range (BR) buffer (Cat#Q33266) and quantified. Samples which were below detection were then processed using the Qubit High Sensitivity (HS) kit (Cat#Q33231) with 2 μl of DNA. For nanodrop readings, 2 μl of DNA from the top, middle, and bottom of the tube was processed on the spectrophotometer. The concentration in ng/μl,

absorbance 260/280, and 260/230 measurements were all recorded. DNA yield was determined by multiplying the total elution volume (100 μl) by the mean DNA concentrations of the top, middle, and bottom of the tube. DNA concentrations were measured with both the Qubit BR kit and Nanodrop, although only the Qubit measurement was used in the final comparison. DNA purity was assessed on the basis of both A260/280 ratios and A260/230 ratios. Again, the mean values from the top, middle, and bottom of the tube were used as endpoints. DNA fragment size can be challenging to measure so we used a three-pronged approach. First, we used the Coefficient of Variation from the DNA concentration measurements obtained from the Qubit BR kit (top, middle, and bottom of tube). If the DNA is very long, it may not be as homogeneously distributed in solution thus one's measurements will be more variable. The coefficient of variation (CV) from repeated DNA concentration measures may indicate the homogeneity of DNA in solution with a high CV associated with long fragments. For the other approaches, we directly measured DNA fragment sizes using an automated pulsed-field capillary electrophoresis system (Femto Pulse Cat# M5330AA, Agilent, Santa Clara, CA) which can reliably measure DNA fragments up to 165 kb in length. We curated two endpoint measures with the Femto pulse; first the mean read length which is automatically generated in the program and second the percent of DNA greater than 50 kb. For the later measurement, we used the 'smear analysis' function built into the Femto pulse analysis program to quantify DNA measurements between 1-10 kb, 10-25 kb, 25-50 kb, 50-100 kb, and greater than 100 kb. These values can be found in the metadata in Supplementary Data 2. To simplify the analyses, we used a final measure of the % of DNA between 50-100 kb and greater than 100 kb.

**Sample exclusion criteria**. Whole blood extracted from nine species of fish was stored at different conditions (95% EtOH 4 °C, 95% EtOH 22 °C, and RNAlater 22 °C) for 0, 1, 3, and 6 weeks (Fig. 1a). Leaf tissue from 4 species of plants was stored in 95% EtOH at 4 °C or RNAlater at 22 °C for 0, 0.17 (4 hr), 2, 7, and 21 days. Of 36 samples, 5 had insufficient DNA (Z. marina RNAlater at 22 °C 2 days, 1 week and 3 weeks; P. torreyi RNAlater 1 week and 3 week) while another 2 samples (Z. marina EtOH at 4 °C, 1 week and 3 weeks) had poor quality DNA thus were excluded from further analysis (Fig. 1a).

**Shallow sequencing**. To assess the influence of sample storage conditions on DNA sequencing, sequencing libraries were made using the newest barcode ligation chemistry (Kit 14) from Oxford Nanopore Technologies (ONT). Samples were processed together to increase throughput and minimize variability during library preparation from pipet shearing of DNA etc. Sequencing libraries were made using either 400 ng (plant) or 800 ng (fish) of DNA as input according to the ONT protocol. The Native Barcoding Kit 96 V14 SQK-NBD114.96 was used to construct libraries either 90 (fish) or 29 (plant) at a time. Each unique pool was sequenced separately on a new Promethion 10.4.1 flow cell. MinKnow version 22.1 was used to process and samples were basecalled using the high accuracy basecalling. At termination of sequencing, libraries were then processed using the NanoPlot tool to generate metrics for each sample including average read length, N50, total reads, total bases, and average read quality[9].

**Evaluation of sample storage on DNA QC and sequencing**. Our goals were to determine if sample storage had an impact on DNA quality and if sample storage had an impact on sequencing results. To ascertain which storage condition (95% EtOH 22 °C, 95% EtOH 4 °C, and RNAlater 22 °C) performed best for fish

samples, we specifically performed a one way Friedman test with repeated measures for the four time points (0 week, 1 week, 3 weeks, and 6 weeks) followed by a multiple comparisons test against the control (0 week, dry ice, $-80\,°C$ stored)[10]. Our design had 4 groups and 9 replicates (species) for the fish samples. For plant samples, we compared 95% EtOH 4 °C, and RNAlater 22 °C across a total of 5 time points (0, 4 hr, 2 days, 7 days, and 3 weeks) and we had 4 replicates (species). We had to use a non-parametric test because sample groups failed to pass normality testing with Shapiro-Wilk[11]. Multiple comparison correction was performed using the two-stage step-up method of Benjamini, Krieger and Yekutieli with an alpha of 0.05[12]. Each of the three storage conditions were independently analyzed to determine if sampling time had an impact. The statistical results could then be compared across the three storage buffer and temperature combinations.

**Impact of short read exclusion kits on sequencing performance.** All 90 fish samples were sequenced with and without a size selection (ONT SFE) (Cat # EXP-SFE001). Specifically, approximately 5 ug of DNA was used as input in a total of 25 µl which was then matched with 25 µl of SFE buffer. This is a slight modification (miniaturization of the protocol which recommends starting with at least 50 µl) and the sequencing read N50 values compared using the Wilcoxon matched-pairs signed rank test (two-tailed) as they did not pass the Shapiro-Wilk normality test ($P = 0.0005$ and $P = 0.0012$). The percent increase in N50 was calculated for the 90 pairs by taking (N50 SFE—N50 noSFE)/(N50 noSFE) * 100. A positive value indicates an increased N50 as a result of using SFE.

**Deep sequencing for evaluation of DNA QC on sequencing and assembly outcomes.** Fish and plant samples chosen for higher coverage whole genome sequencing were additionally size selected using ONT size selection (Oxford Nanopore Technologies, Cat# EXP-SFE001) to deplete short fragments under 25 kb. We performed deep sequencing on a subset of plant and fish samples to evaluate assembly metrics. For the three fish *K. azureus, M. californiensis*, and *G. nigricans*, we sequenced samples stored at 95% EtOH at 22 C for 6 weeks. For *K azureus*, we also sequenced the control and the sample stored in RNAlater at 22 °C for 6 weeks. For plants, we sequenced the samples stored in 95% EtOH at 4 °C for 3 weeks from two important food crops, *M. esculenta* (cassava) and *S. bicolor* (sorghum). For cassava, we also sequenced the control and the samples stored in RNAlater at 22 °C for 3 weeks (Table 1). For assembly validation, we followed the Vertebrate Genome Project (VGP) guidelines[5]. Data quality control was performed with FastQC version 0.11.0 and NanoPlot 1.40.0[9]. Unassembled genome analysis and size estimation was completed using GenomeScope 2.0[13]. Samples were assembled using Flye version 2.9 with ONT reads[14]. All assemblies were polished with the raw ONT reads using Racon version 1.4.20[15]. All assemblies were additionally polished with the associated Illumina WGS reads using Pilon version 1.24 after the initial Flye assembly and Racon polishing[16]. Both Racon and Pilon polishing used Minimap2 version 2.21[17]. Final assembly metrics were calculated using assembly-stats version 1.0.1 and completeness was estimated using BUSCO version 5.4.3[18]. The BUSCO databases eudicots_odb10, liliopsida_odb10, and actinopterygii_odb10 were used for cassava, sorghum, and fish samples, respectively. QV was calculated using Meryl version 1.3 and Merqury version 1.3[8].

**Methylation frequency.** Genomic reads from the raw nanopore FAST5s generated from *M. esculenta* and *K. azureus* deep sequencing samples were used for methylation calling. Genome

assemblies generated for the same individuals were used as references for alignment. FAST5 data were converted to POD5 format using the pod5 software package (https://github.com/nanoporetech/pod5-file-format). Methylation calling was performed with ONT basecalling software Dorado version 0.2.4 (https://github.com/nanoporetech/dorado/). Dorado uses the raw POD5 data and a reference to identify methylated cytosines. This was performed with the super high accuracy (SUP) base calling model trained for R10.4.1 chemistry and 260 bps translocation speed, matching the sequencing conditions. The assembled genomes generated from each sample were used as references to generate an aligned BAM file with MM/ML tags containing 5mC and 5hmC methylation calls. These were then piled up with modkit (https://github.com/nanoporetech/modkit), and the piled-up calls (aggregating 5mC with 5hmC) were used for calculating genome-wide methylation frequencies across all CG sites.

**Statistics and reproducibility.** Statistical analyses and figures were made either using R or Prism version 9.4.1 with a paid license. Throughout this study, biological replicates were defined as fish species ($n = 9$) and plant species ($n = 4$). This has the benefit of demonstrating reproducibility across multiple species, while acknowledging the drawback of not including an additional level of replication at the species level, which would've been cost-prohibitive. We believe the sample size sufficient to describe marine fish blood due to consistent results, and relatively sufficient for plant leaf tissue where results were less consistent across species. Further studies using nucleated blood from other vertebrates, or with plant tissues beyond leaf, will be necessary to reproduce this method beyond the tested sample types. But the results are encouraging. To support reproducibility of sampling and DNA extraction, we have supplied Standard Operating Procedures for fish and plants as Supplementary Notes 1 and 2, and the sequencing parameters are translatable across any Nanopore instrument.

**Reporting summary.** Further information on research design is available in the Nature Portfolio Reporting Summary linked to this article.

## Data availability

All DNA QC and shallow sequencing QC data are represented in Supplementary Data 2. Sequencing data is publicly available (PRJNA971989). We've included two standard operating procedures (SOPs) for sample collection, storage, and extraction as supplementary data. This includes one for fish (Supplementary Note 1) and one for plants (Supplementary Note 2).

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

## Acknowledgements
This work was funded by a Bill and Melinda Gates Foundation (BMGF) grant (INV-040541) to T.P.M., National Science Foundation (NSF) graduate research fellowship (Fellow #2021321499) to M.L.M. and NSF Postdoctoral Fellowship in Biology "Rules of Life" (Award #2011004) fellowship to J.J.M. We thank the Michael lab for helpful conversations and input into this study. We thank John Tyson's work on developing aspects of the short read elimination methodology and sharing on twitter. We also thank the San Diego Botanical Gardens and Salk greenhouses for donating plant tissue.

## Author contributions
J.J.M., M.L.M., and T.P.M. conceived and designed the experiments. J.J.M. carried out the collections, molecular biology, sequencing, and analysis of the fish samples. M.L.M. carried out the collection, molecular biology, sequencing, and analysis of the plant samples. J.J.M. and M.L.M. wrote the manuscript. N.A.A. performed the assemblies of fish and plant samples. A.A. led methylation basecalling and analysis. E.R.M. collected and aggregated data on Femto Pulse. L.T. assisted in molecular biology (extractions from fish tissue). T.P.M supervised the project.

## Competing interests
The authors declare no competing interests.

## Ethics
No human data was used in this study. Animal samples were donated by local fishers and sample processing under IACUC Animal Use Protocol S12219.
