## [Peer review file · Communications Biology]

REVIEWERS' COMMENTS:

Reviewer #2 (Remarks to the Author):

In this manuscript, the authors conducted an in-depth investigation into the impact of different sample handling conditions on the quality of DNA extraction in fish and plant samples. The standardized method proposed by the authors seems innovative, especially for the current genomics research. The method is highly relevant and timing. Standardizing sample preparation methods helps reduce experimental errors and is crucial for ensuring the repeatability and accuracy of research, particularly in genomics studies that deal with a large amount of genetic information. Therefore, the method proposed in this paper can support for future genomics research.

However, it should be noted that the 10 species used in this study may not represent the universality of the method, because samples from different environments and from different tissues/organs can affect preservation quality. For example, in plants, it is relatively easy to extract nucleic acids from young leaf tissues, but mature parts such as lignified tissues are difficult to extract and preserve. Therefore, to increase the general applicability of this method to samples from different taxa, growth environments, or tissue types, it seems required to take further research to address these challenges, which would make the method more meaningful. I agree to accept this manuscript for publication after minor revisions. The main concerns are:

1. There is one fish species, *Halichoeres semicinctus*, for which no relevant results were obtained due to insufficient sample quantity. Therefore, it is advisable to remove it from the article to avoid confusion.
2. The format of Supplementary Table 1 is not very clear and may be difficult to understand. It is necessary to transform the table into a more readable and understandable format.
3. Figure 1 is rather insufficient, as it lacks clear separations, making it appear cluttered. It is advisable to use different colored boxes or blocks to improve the clarity and readability of the figures.

Reviewer #3 (Remarks to the Author):

I have gone through the previous reviewer comments and think that they have been answered sufficiently. I have issues to identify the "swapped order" of figures 3 and 4 as mentioned by the authors (this might apply to supplementary figures 2 and 3). And I do not understand why they present all results for plants in one figure, but split results for fishes into two figures.

Overall, I find the manuscript interesting, but not providing much news, except that degradation of samples in EtOH is tested by sequencing and assembly. The paper is not telling us much, if non-sequencing based methods are sufficient to predict the outcome of sequencing. That would have been an interesting analysis (there is just one sentence hinting at Suppl. Table3 which looks like unformatted R output).

It seems to me this study is the byproduct of another study. I agree with reviewer 2 that this manuscript is more suitable to a methodological oriented journal, and I am not sure if it is the scope of Communications Biology.